# Importance of Human Factors in Text-To-Speech Evaluations

*Lev Finkelstein,  Josh Camp,  Rob Clark*

Google
{finklev, joshcamp, rajclarck}@google.com

## Abstract

Both mean opinion score (MOS) evaluations and preference tests in text-to-speech are often associated with high rating variance. In this paper we investigate two important factors that affect that variance. One factor is that the variance is coming from how raters are picked for a specific test, and another is the dynamic behavior of individual raters across time.

This paper increases the awareness of these issues when designing an evaluation experiment, since the standard confidence interval on the test level cannot incorporate the variance associated with these two factors. We show the impact of the two sources of variance and how they can be mitigated. We demonstrate that simple improvements in experiment design such as using a smaller number of rating tasks per rater can significantly improve the experiment confidence intervals / reproducibility with no extra cost.

**Index Terms**: Text-to-Speech, subjective evaluations, MOS, preference tests, comparative tests, test reproducibility

## 1. Introduction

Both mean opinion score (MOS) evaluations and preference tests in text-to-speech (TTS) often use a standard confidence interval. It is perceived as a sufficient safeguard for the rating variance, while, as we show in this paper, it lacks the ability to address a few major factors contributing to this variance.

We investigate two important factors that affect the variance. One factor is that the variance is coming from how we pick the raters for a specific test. Namely, besides the objective audio sample quality, each rater has their own preferences, so if the rater pool is large enough (e.g., in crowdsourcing projects), and the actual number of raters chosen for a specific test is too small, then test reproducibility will become a challenge since averaging over the scoring tasks will not remove the bias associated with raters' personal preference.

MOS evaluations require scoring an individual audio sample, this is a less constrained task than a two-sided comparison in preference tests, so intuitively raters variance can be more affected here. Note that this kind of variance is something that cannot be seen from the test scores alone, so it is not directly reflected in the confidence intervals. More elaborate techniques such as intra-class correlation analysis can help to get better estimations, but they also require a sufficient number of data points, as well as a proper experiment design to be used effectively.

An additional variance factor is the order-dependence of sequences of ratings. If a rater rates multiple samples sequentially, then their rating for a given item technically depends on its position in the sequence and is not independent of their other ratings. We do not know how a particular rater will behave a pri-

ori – this may be a result of some inherent calibration process (learning curve) of each rater, or fatigue for a large number of rating tasks. In any case, this phenomenon can definitely affect the ratings both in MOS and in preference tests.

In this paper we demonstrate the impact of these two sources of variance. The rater-induced variance impact is evaluated by bootstrap analysis [1], and the impact of the order-dependent ratings is analyzed by showing non-random trends in the score behavior. Intuitively, the first issue may be mitigated by introducing more raters for a specific test, and the second factor by limiting the number of audio samples per rater, and we show this is indeed the case. Using 60 audio samples per rater instead of 10 may double the variance even if the number of rating tasks remains the same. While it is a natural decision, it is not always taken into account in the design of this type of the experiments.

## 2. Related work

The issue of score variation arising from rater sampling has been studied previously. In [2], the authors show that per-participant MOS values vary considerably within the same test, and in an analysis of the 2013 Blizzard Challenge results [3], the authors found the number of raters to be a key factor in test reliability and sensitivity. However, neither study attempts to characterize the variance of the sampling distribution directly as is done in the present work.

The influence of the number of ratings completed by a single rater has, to the best of our knowledge, not been studied in the context of subjective evaluations of TTS systems. The issue has received attention in the context of crowdsourced evaluations of degraded speech, however (e.g. those observed in telephony). In [4], a study was conducted in which crowdsourced workers completed an MOS evaluation of degraded speech samples from [5], with three groups rating 10, 20, and 40 samples respectively. They found that while the groups that rated 10 and 20 samples performed similarly, the group that rated 40 samples reported much higher levels of fatigue, and had lower participant retention. For the 40-sample group, they also found rater performance (as measured by correlation with laboratory results) to increase throughout the first half of the samples and decrease in the second half. Contrarily, in a study of crowdsourced spoken word recognition, authors of [6] found rater performance to improve in the second half of the task, which they attribute to increased familiarity with the task. It seems there may be competing factors at play: as the number of rating tasks increases, performance improves, but so does fatigue. Word recognition, however, is presumably less subjective than TTS evaluations.

In this paper we show that the same calibration and/or fa-

tigue phenomena present in TTS subjective evaluations as well, and that it causes a clear monotonic trend both in MOS and in preference test outcomes. Unlike [4], where the benchmarks were either self-reported fatigue scores and correlation with laboratory results, our results demonstrate the influence of number of ratings intrinsically.

Reliability of judgments is a known problem, and some methods, including using intra-class correlation coefficient, may help in the analysis (see, for example, [7, 8]), but these methods require a specific experimental design. An application of cluster-based methods to text-to-speech tasks was done in [9], that used them for evaluating both MOS and preference tests. In particular, it was observed that the number of listeners has a strong impact on the confidence intervals (a fact that is often ignored if using out-of-the-box methods for confidence interval estimation), and that MOS tests are more sensitive to the number of listeners than preference tests.

## 3. Evaluating rater distribution impact

We have two independent methods for evaluating the variance associated with these human-related factors. First, we use a bootstrap-like methodology to estimate the impact of the rater distribution. Second, we perform a special time-based analysis to investigate dynamic rater behavior.

### 3.1. Formal setup

We start with the rater distribution evaluation. To evaluate the impact of rater distributions, we investigate the reproducibility of the test scores at the test level.

Let us define an MOS experiment setup[1] $\mathcal{M}$ as a mapping from a set of audio samples $\mathcal{S}$ and rater pool $\mathcal{R}$ to the MOS. We may assume that such a mapping depends on the rater distribution, on the instructions presented to the raters, and on the way the samples are assigned to the raters. So, we can assume an existence of some distribution of MOS scores, $P_{\mathcal{S},\mathcal{R}}(\mathcal{M})$ that describes applying a setup $\mathcal{M}$ for the same set of audio samples $\mathcal{S}$ and the same rater pool $\mathcal{R}$.

We may measure the variance of the rater-associated factors by measuring the deviation of the distribution above given the rest of the factors, such as the instructions and specific samples to be tested, which are not affected. Note that directly measuring of the variance by rerunning the same test many times is very resource-consuming, due to the distribution of the standard deviation. Instead, we use an approach which is a variation of bootstrapping.

More specifically, we created a large test with $N$ audio samples, and required each sample to be evaluated by $L$ different raters. After that, we are able to randomly sample one rating per item under certain constraints (such as a fixed number of samples per rater), thus creating a simulated test[2]. This simulated test can then may be used to estimate the per-test score distribution under these constraints, without running a large number of real experiments.

Formally, if the real test $\mathcal{T}$ contains ratings $R_{ij}$, where $i$ is the audio sample number and $j$ is the rating index of this item, a simulated test $\mathcal{T}_n$ is a subset of $R_{ij'}$ of $R_{ij}$, where each $i$ appears exactly once, and $j'$ is a single rating among the available ones. The average score (MOS score) of such a simulated test is

$S_n = \text{Mean}(R_{ij'})$, and the standard deviation among $S_n$ can be used a reliable estimation of the test-level deviation for the test using exactly one rating per item.

Note that in order to simulate a test with up to $K$ samples per rater and 1 rating per sample, we need a special sampling procedure. Ideally, 1000 samples with one rating per sample and the limit of up to 60 stimuli per rater should require 17 raters (e.g., 16 raters with 60 stimuli each, and one rater with 40 stimuli). If sampling for bootstrapping purposes is performed in a random order, however, such dense packing will probably not be achieved since the same stimuli are rated by a number of raters, which may create scheduling conflicts. A naive sampling could result, for example, in associating 33 raters with 30 stimuli each, and one rater with 10 stimuli. In order to simulate a dense schedule, we implemented a greedy scheduler that minimizes the number of raters given constraints. In practice, even a greedy scheduler cannot obtain the optimal dense packing since the data available for bootstrapping is limited, and there still will be slightly more raters participating in each simulated test than we could theoretically get in real life.

### 3.2. Experimentation setup

We created two large tests of 990 audio samples. The samples, a few seconds each, were generated by a TTS system using a 24K sample rate. The tests were crowdsourced, with 10 ratings per sample, where the samples were given to the raters in batches of 10 samples. In one test each rater was allowed to evaluate up to 60 samples, while in another test each rater was only allowed to evaluate 10 samples. A histogram of the actual number of samples per rater in the first test is shown in Figure 1. Note that not all raters completed rating 60 samples. The second test had a limit of up to 10 samples per rater, so, given the samples were presented in batches of 10, each rater rated exactly 10 samples.

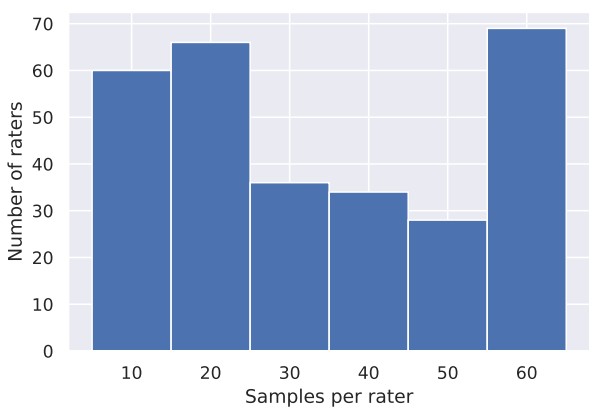

Figure 1: *Histogram of samples per rater in the first test (up to 60 samples per rater).*

### 3.3. Experiments: Robustness

We analyzed the standard deviation for both tests using the bootstrapping methodology described above. We generated 1,000 simulated tests from our real data identical to the real test conditions, but with a single rating per sample, and calculated their MOS scores. The graph of the standard deviation for these scores as a function of the maximum number of samples per rater is shown in Figure 2 (top). Note that the standard devia-

---

[1]This set of definitions is for the MOS tests, but it can be applied to the preference tests as well.

[2]We experimented with multiple ratings per sample with similar results, so we use this setup for simplicity.

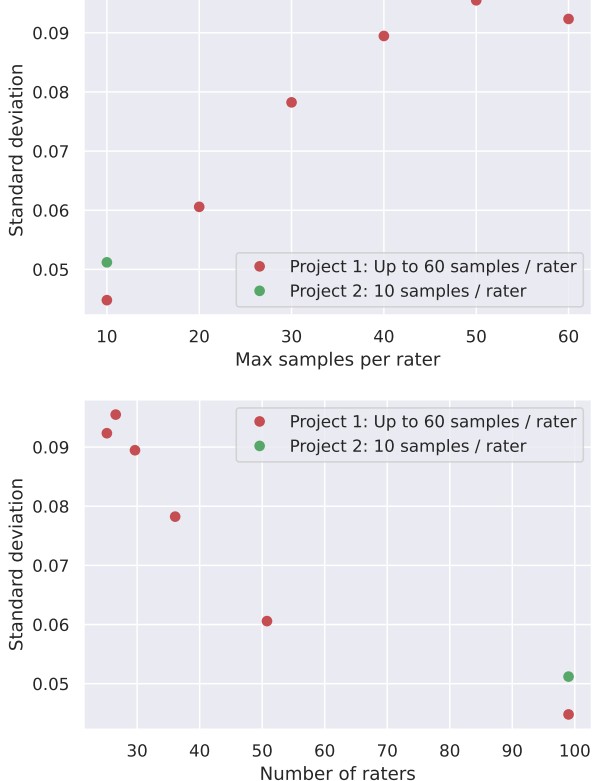

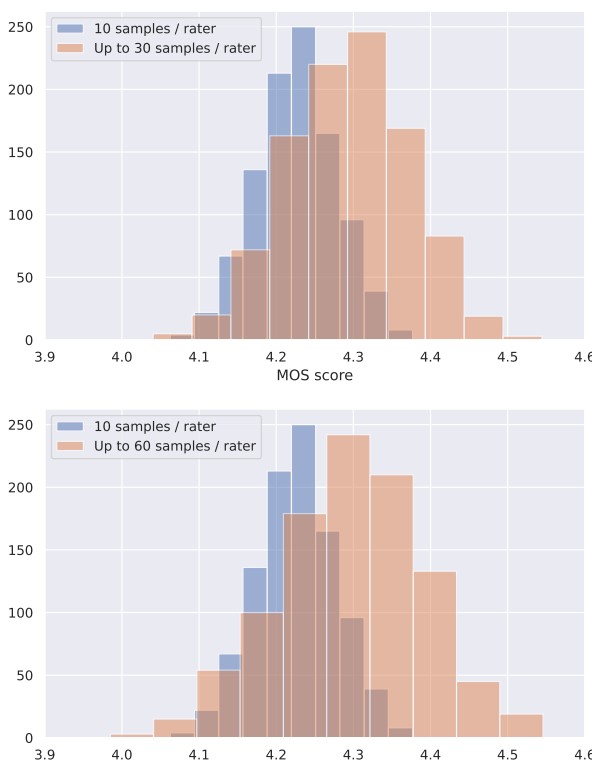

Figure 2: *(Top) the standard deviation as a function of max samples per rater, (Bottom) the standard deviation as a function of the number of raters in the simulated tests.*

Figure 3: *Histogram of MOS scores of simulated tests. (Top) K = 10 vs. K = 30 samples per rater, (Bottom) K = 10 vs. K = 60 samples per rater.*

tion number increase is almost by factor of 2, which means that the confidence interval should be doubled.

The same plot contains a single green dot corresponding to the (real) test with 10 samples per rater. The deviation for this setup is higher since the simulated data for that test was sampled from the artificial pool of the second test with many more raters, vs. the artificial smaller pool of the first test, thus leading to a higher variance. Note that the artificial scheduling in this framework is not really capable of getting real cases like 16 raters with 60 samples per rater since the data available for bootstrapping is limited, so the deviation in our graph is presumably lower than in real life. The dependency of the standard deviation on the number of raters in the simulated tests is shown in Figure 2 (bottom). It is possible to see that simulated sample limits of 40, 50, and 60 samples per rater resulted in a very close number of raters.

### 3.4. Experiments: Distribution

From the distribution point of view, the difference may be viewed in Figure 3 showing the histogram of MOS scores of the simulated tests. Each test has a limitation of $K$ samples per rater. The top graph corresponds to $K = 10$ vs. $K = 30$, and the bottom graph to $K = 10$ vs. $K = 60$. Having 10 samples per rater significantly reduces the deviation of MOS scores, presumably due to a larger number of raters.

Another interesting observation is that the number of samples per rater leads to an increased MOS score. The experimen-

tation includes many raters, so it should not be a random fluctuation. We also observed similar behavior in other experiments not mentioned in this paper. We don't know the exact reason for the difference in the average. It is possible that this is an artifact of a rater getting assigned a large number of successive rating tasks, which leads to some kind of a bias as discussed in section 4.

### 3.5. Experiments: Different speakers

Intuitively, different voices may have a different score variance. To analyze the behavior of the standard deviation as a function of number of samples for different voices, we compared the behavior of two different speakers (60 samples per rater both), where the quality of the first speaker is better. Note that the standard deviation depends on the MOS scale, so to present the speakers on the same scale, we multiplied the deviation of the second speaker by the coefficient equal to MOS(first speaker) / MOS(second speaker). The results are shown in Figure 4. We hypothesize that a higher variance of the second speaker is caused by the fact that their voice quality is worse, thus leading to a wider MOS dynamic range.

## 4. Evaluating the impact of order-dependent ratings

In this section we show how to validate the impact of the order-dependent scores of the raters that get more than one rating task. It is interesting that not all the experiments are subject to this

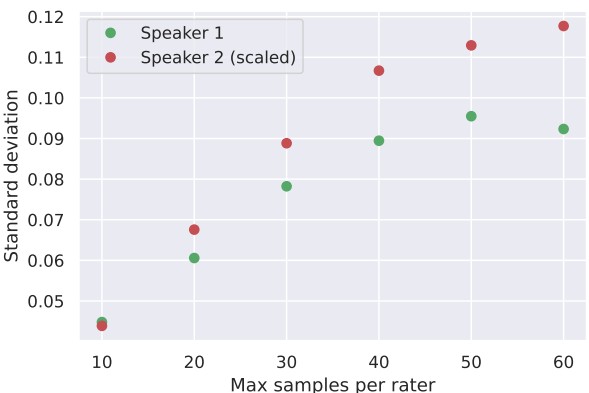

Figure 4: *The standard deviation as a function of the number of raters in the simulated tests for two speakers.*

change in ratings. Also, the impact, if present, is not necessarily positive or negative; we saw experiments with both trends.

Assume that test $\mathcal{T}$ (either MOS or preference test) contains scores $S_{rj}$, where $r$ is a rater, and $j$ is the serial index of the sample obtained by this rater[3]; in other words, $S_{rj}$ represents the $j$th rating obtained from rater $r$. Let us select a set of raters $\mathcal{R}$ having at least $K$ ratings. Then we can define a special value $S_{\mathcal{R}}(k)$ with $k \leq K$ as the cumulative average over all the rating tasks with the serial number of $k$ or less:

$$S_{\mathcal{R}}(k) = \frac{1}{k|\mathcal{R}|} \sum_{r \in \mathcal{R}, j \leq k} S_{rj}. \qquad (1)$$

We use the cumulative average since it better reflects the dynamics of the number of samples per rater. Note that we needed to preselect the set of raters $\mathcal{R}$ to have at least $K$ ratings in order to have the same rater population in every slicing. If the sample ratings were independent, the behavior of $S_{\mathcal{R}}(k)$ as a function of $k$ would be more or less random and have no clear monotonic trends. In our experiments, however, we demonstrate that the behavior is often systematic, with relatively long monotonic regions. In the next sections we show the behavior of $S_{\mathcal{R}}(k)$ in different setups. We also present a special sample-based analysis to prove our hypothesis using a different metric.

### 4.1. Raters' scores in preference tests

In the first experiment we analyze the ratings in two different preference tests. The preference test setup we use is actually a comparative MOS (CMOS) task where raters score the sample on the whole-number scale of -3 to +3, where -3 is a strong preference for one stimulus and +3 a strong preference for the other. The raters were able to rate up to 60 samples. Note that not all raters will achieve this. The cumulative average $S_{\mathcal{R}}(k)$ as a function of k for the raters that rated at least 40 and at least 60 samples is shown in Figure 5. Intuitively, a monotonic trend in both graphs after about 10 samples should not be random, but we cannot conclude it from the graph only, and a more formal analysis is given in Section 4.3. Also in both these cases the average scores are all positive, reflecting that the experiment voice turned out to be considered better than the baseline.

[3]The notation here differs from Section 3.

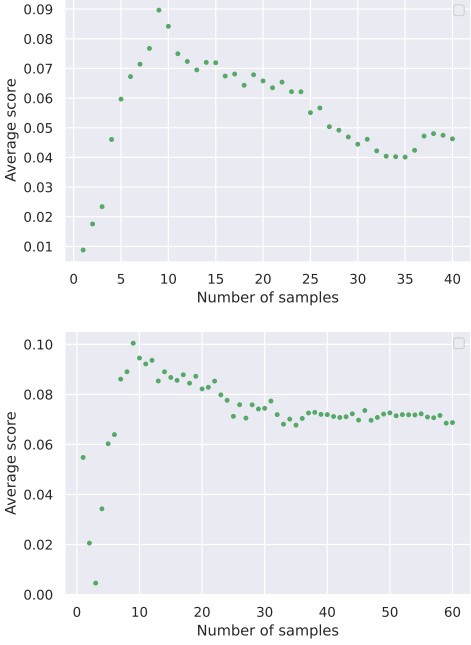

Figure 5: *The average score as a function of the number of samples, for the raters that rated at least 40 samples (top), for the raters that rated at least 60 samples (bottom).*

### 4.2. Raters' scores in MOS tests

The MOS tests are subject to the same phenomenon. We compared the MOS tests for two speakers[4]. Each test used 10 ratings per sample, with up to 60 samples per rater in batches of 10. The results are shown in Figure 6. We do see a clear trend in the second speaker test but not in the first speaker test. The first speaker has a higher quality, so it is possible that fatigue / calibration plays a lesser role than for the second speaker.

### 4.3. Sample-level analysis

Since there is still a chance of monotonic trends occurring in random sequences, we performed a different type of analysis to support the existence of the fatigue / calibration trend. In this series of experiments we analyze tests with multiple ratings per audio sample and show that the ratings have a monotonic trend.

Let us have a test with $L$ ratings per audio sample, and let $T(r, X)$ be a rating task of rater $r$ associated with sample $X$, and $S_r(X)$ be its score. Assume that for each rater $r$ we sort all the rating tasks $\{T(r, X)\}$ performed by this rater in time-based order, such that each task becomes associated with the corresponding ordinal number from 1 to $|\{T(r, X)\}|$, which we denote by $N(r, X)$. For example, $N(r, X) = 2$ means that sample $X$ was the second audio sample rated by rater $r$.

Let $\{S_r(X)\}$ be the set of multiple scores of the same audio sample $X$, and assume that we define an order on this set, based on $N(r, X)$. Namely, we say that $S_{r_1}(X) \preceq S_{r_2}(X)$ iff $N(r_1, X) \leq N(r_2, X)$. Let $V(X)$ be the vector obtained by sorting $\{S_r(X)\}$ according to the relation above. Each vector $V(X)$ contains $L$ items (the number of ratings per sample), and due to the nature of the relation, if $i < j$, then the rating $V_i(X)$ is associated with the "earlier" rating than $V_j(X)$ (not necessar-

[4]The two tests from Section 3.5.

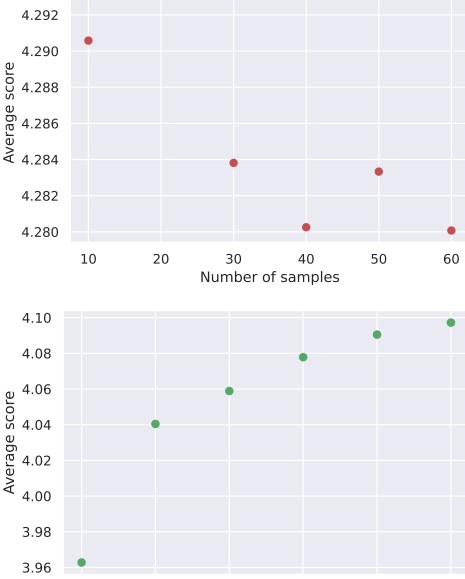

Figure 6: *The average score as a function of the number of samples, for the raters that rated at least 60 samples, for two different speakers.*

ily of the same rater), since the relation promotes early ratings of each rater.

We may, therefore, define $L$ artificial evaluation tests $\mathcal{T}_i = \{V_i(X)\}$, each one containing the $i$-th slice across all vectors $V(X)$. The average $M_i = \text{Mean}[V_i(X)]$ is an average evaluation score (e.g., MOS or CMOS) of $\mathcal{T}_i$, where smaller indices are associated with earlier ratings. Note that since the order of the elements in $V(X)$ is not uniquely defined for the case $N(r_1, X) = N(r_2, X)$, $\mathcal{T}_i$ are not uniquely defined either. To avoid random fluctuations, we used the averaged values $M_i' = \text{Mean}[M_i]$ over large number of random iterations.

In the rest of this section we show that the resulting vector $(M_1', M_2', \ldots, M_L')$ has a clear monotonic trend, at least when the number of rating tasks per rater is large enough. We use the Mann-Kendall test [10,11] to validate the monotonicity hypothesis. The implementation is based on [12] for a small number of data points. We calculate the Mann-Kendall statistic

$$S = \sum_{k=1}^{L-1} \sum_{j=k+1}^{L} sign(M_j' - M_k'),$$

and look for the $p$-value for the null-hypothesis of no trend to $S$ and $L$ in the table specified in [12]. The lower $p$-value is, the more we are confident that the sequence is monotonic.

Table 1 shows the outcome for different tests and different configurations. The tests were conducted on different sets of audio samples and required 10 ratings per item, except $\mathcal{T}_9$ and $\mathcal{T}_{10}$ with 8 ratings per item. It is possible to see that all the tests with many tasks per rater had a clear monotonic behavior (except maybe $\mathcal{T}_1$ which had a somewhat high probability threshold). However, tests with a small number of tasks per rater had a less clear behavior – some of them had a monotonic behavior, while some didn't. Other parameters like the nature of the test (MOS / CMOS) seemed to have no impact. Neither

we were able to predict whether the sequence of $M_i'$ is increasing or decreasing. We hypothesize that there are two trends, calibration and fatigue, where the calibration trend affects some of the raters even for a small number of rating tasks, while the fatigue trend affects almost all the raters having a large number of tasks.

Table 1: *Mann-Kendall $p$-value and monotonicity trend for different evaluation tests.*

| Test | Type | Samples per rater | Trend | p-value |
|------|------|-------------------|-------|---------|
| $\mathcal{T}_1$ | MOS | 60 | Down | 0.108 |
| $\mathcal{T}_2$ | MOS | 60 | Up | 0.014 |
| $\mathcal{T}_3$ | MOS | 60 | Up | $< 0.001$ |
| $\mathcal{T}_4$ | MOS | 60 | Up | 0.023 |
| $\mathcal{T}_5$ | CMOS | 60 | Down | 0.014 |
| $\mathcal{T}_6$ | CMOS | 10 | Down | 0.431 |
| $\mathcal{T}_7$ | MOS | 10 | Up | $< 0.001$ |
| $\mathcal{T}_8$ | MOS | 10 | Up | 0.431 |
| $\mathcal{T}_9$ | MOS | 6 | Up | 0.500 |
| $\mathcal{T}_{10}$ | MOS | 7 | Down | 0.031 |
| $\mathcal{T}_{11}$ | CMOS | 5 | Up | 0.014 |

## 5. Discussion

This work has focused on two very significant sources of variability in the TTS evaluations. The first one, caused by the variance among raters, may be considered well-proven, but taking it into account in a confidence interval requires a more elaborated setup than is typically used in TTS evaluations. However, it can be addressed by increasing the number of raters, which leads to reducing the variance without affecting the number of rated samples.

In our experiments we observed a substantial improvement by increasing the number of raters to a rather large number. This corresponds to the findings in [3], where the recommendations were to use about 30 paid raters in controlled conditions, and many more raters (the exact number was not specified) for less controlled scenarios like crowdsourcing.

An interesting question is whether this behavior is common for all MOS tasks (text-to-speech synthesis, voice conversion, speech enhancement, etc.). We would expect some difference since we observed the difference even across the samples produced by the same TTS system for different speakers (see Section 3.5). We believe though that the variance associated with the rater choice should be inherent to MOS tests, thus creating a similar type of the dependency on the number of raters, even if the absolute numbers differ.

The second factor that is analyzed in this paper is caused by a dynamic trend in the raters' rating process. This factor is more vague. While we observe its existence, we cannot claim exactly what the source of this type of behavior is—fatigue, or some process of raters self-calibration, or something else. It is also unclear how different this factor is for different raters. It is possible that this type of problem may be mitigated by modifying the instructions for raters, in a way to keep them more alert and calibrated.

Note that the tradeoff between calibration and fatigue is hard to analyze given the lack of ground truth in this type of evaluation. So, we assume that there should be a minimal number of audio samples for the calibration, but the paper doesn't

set a goal to find this number (and it is unclear if it is feasible in the current setup). Given that reducing the number of tasks per rater also requires more raters and thus reduces the impact of per-rater variability (the first factor), we do consider limiting the number of tasks per rater beneficial.

## 6. Conclusions

In this work we presented the analysis of two important aspects of TTS evaluations that are currently not taken into account by the way confidence intervals are usually calculated.

The first factor is caused by the rater variance, i.e. by picking the raters from the rater pool. We showed the impact of this type of variance using bootstrapping simulations on tests with a large number of raters and with multiple ratings per task. In particular, we showed that using 60 audio samples per rater instead of 10 may double the variance even if the number of rating tasks remains the same.

The second aspect implies that we have a non-random component in our evaluations that depends on the number of tasks performed by the rater, which causes the scores to behave monotonically depending on the order of the rating task. While this does not necessarily increase the variance, this factor leads to quality-unrelated scoring of items, and affects both MOS and preference tests. It is unclear though whether all raters are subject to such a behavior, or only some of them. Our results demonstrate the presence and the impact of this phenomenon intrinsically. Increasing the number of raters, which is equivalent to reducing the number of rating tasks per rater, helps to partially mitigate the problem.

It is difficult to give recommendations regarding the exact number of raters since the process is affected by many factors. There is a tradeoff between the necessity for a rater to learn the task on one hand and not to be affected by fatigue on the another. In this paper, we used the minimal number of 10 audio samples per rater, and the number of raters was derived from the number of samples per rater, e.g., a test with 1000 audio samples required 100 raters. For a different type of the test, it may be beneficial to fine tune these numbers by calculating confidence intervals using the techniques like [9] that take into account the rater variability.

Increasing awareness of these factors will allow researchers to make more informed decisions when setting up TTS evaluations. A large number of rating tasks per rater may lead to the evaluation artifacts that are typically not addressed in the way results are analyzed, and the results of such experiments may not be reproducible. However, very simple changes in experiment design may significantly improve the reproducibility (and potentially provide a more precise score) without changing to the number of overall rated items.

## 7. Acknowledgments

The authors would like to thank Tilman Achberger for his valuable comments on the experiment design, and to the reviewers whose input helped to make the experiment description and the discussion much more accurate.

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
