# OpenReview forum: "Importance of Human Factors in Text-To-Speech Evaluations"
_Interspeech.org/2023/Workshop/SSW — SSW12_

### Official Review · Reviewer_45qK · 2023-06-02
**Please clarify the simulation settings in more detail.**

**Rating:** 7
**Confidence:** 3

**Review:**

## Key Strength of the paper
This work analyzes important human factors of the rating variances in subjective evaluations.
Authors hypothesize that these factors derive from raters' preference (bias) and order-dependency of ratings.
To try to reveal their influences, authors have conducted simulated evaluations with some conditions and discussed them in details.

## Main Weakness of the paper
- This paper does not describe several information about evaluation setup such as:
  - audio information: corpus name, sampling rate, sound type (recorded sound, TTS system), average duration of stimuli in each sample set
  - situation, crowdsourcing or on site, room information (e.g., soundproof room, general room)
  - equipment: speaker or headphone and its type
  These conditions affect evaluation results.

## Novelty/Originality
Authors' challenges are original.

## Suggestions for improvement
- Please describe additional evaluation setup mentioned in "Main weakness of the paper" section.
- Please add index terms.

## Quality of References
Authors refer older and newer papers properly.

## Clarity of Presentation
I can understand contents in this paper.

---

> ### Author Response · Authors · 2023-06-28
> **Thank you very much for your review!**
>
> This is the summary of what we did to address your comments:
>
> > This paper does not describe several information about evaluation setup …
>
> Description updated (there were crowdsourced projects, no specific control on the environment).
>
> > Please describe additional evaluation setup mentioned in "Main weakness of the paper" section.
>
> Done
>
> > Please add index terms.
>
> Thank you very much, those were omitted by mistake.

---

### Official Review · Reviewer_owVx · 2023-06-05
**Interesting research on implications of some design aspects of MOS/preference test.**

**Rating:** 7
**Confidence:** 4

**Review:**

This paper analyses how two human factors affect the variance of MOS/preference test evaluations. 1) rater bias -each people may have different views; 2) variation of rates with time (learning vs fatigue). This is a very interesting topic as human evaluation is still dominant in our area and reducing the variance would allow to learn and progress faster.

Based on an analysis on a large evaluation the authors give a clear recommendation: increase the number of raters reducing number of samples per rater. However, it is not clear if this is a general recommendation or specific for their particular evaluation. Furthermore the clarity of the paper could be improved.

The paper present some facts and give a recommendation, but I think the authors could be more precise.
The authors recommend to increase the number of raters to reduce variance, but they report different results in reference [3]. The justification is that technology know is different and that people are more exposed to TTS. As a practitioner, I am not sure if the recommendation is valid for my technology, or should I do similar test? What about using MOS for e.g. voice conversion or speech enhancement?

The paper does not explain the technology used to generate the samples, but I think it should be explained. E.g.: how many systems are the raters exposed? I would say 1 TTS, not sure if one or two voices. Does the number of systems influence the results? E.g in Blizzard Challenges (ref [3])  many systems participated and maybe this helped raters did a self-calibration? Another possible explanation: maybe if you only listen to one system after scoring a few similar rates (4, 4, 4) raters try to put attention to small differences to be able to provide valuable feedback. However, if several systems are presented and the rates have more natural differences, the raters feel a ‘4’ for a given system is all it’s needed as they use different scores for other systems.

To summarise: thanks to the authors for pointing out the differences with reference [3], but in my opinion further evidence is required to give such a recommendation.

On the other hand, in case the authors justify that the recommendation is valid for a wide range of tasks and technologies, I think it would be useful to make explicit the benefit (e.g. translate score+variance into confidence ranges) and they could give a specific recommendation (e.g. as much raters as possible? or number of ratings per rater < K (k=10?).

The conclusion for the second factor analysed  (temporal evolution of ratings) is less clear. Table 1 shows that sometimes goes up, sometimes goes down. The authors give two possible causes: learning vs fatigue. However in conclusions the recommendation is to reduce number of rating tasks per rater. I am not sure this is generally good recommendation (it’s for some cases, not for others: helps with fatigue, does not with respect learning the task).

About the clarity:
- The authors mention in several parts of papers (e.g. 3rd paragraph of introduction) that the variance can be analyzed using “intra-class” correlation. I would appreciate a reference to understand what they are referring. This is again mentioned in end of section 2 (this time with references) and in the discussion section “a more elaborated setup than is typically used”. It’s clear that the authors consider it a good solution but more complex and I recommend to explain a little bit so that the readers can be convinced that it’s not worth the effort of the complex setup compared with the simple recommendation in the paper.
- In section 3.1, I am afraid I don’t understand what is the sampling procedure, or more specifically what is “schedule” in this context.  “the schedule containing 16 raters with 60 stimuli and one rater with 40 stimuli should be preferred over a schedule containing 33 raters with 30 stimuli each and one rater with 10 stimuli. For this purpose, we implemented a simple greedy scheduler that minimizes the number of raters given constraints”. Why are 16+1 rater preferred to 30+1? Is schedule a good word with respect “sampling criterium”?
- In section 3.2, I would add some information. As I already mentioned, please explain how samples are generated. But also, indicated for both test how many raters and mean of rating per rater (from figure, around 240 ratees, total ratings 7400, approx 7 rates per sample?). What about second test?
-  Maybe I don’t understand the sampling, because I don’t understand this sentence: “so, there are actually more raters participating in each simulated test than we could theoretically get in real life”

---

> ### Author Response · Authors · 2023-06-28
> **Thank you very much for your review!**
>
> This is the summary of what we did to address your comments:
>
> > Re difference with [3]:
>
> You’re absolutely right, we mentioned the familiarity with TTS as one of the factors that we considered the most possible (we had no space for more). After your comment, we analyzed the difference in the setups more carefully. The number of 30 listeners in [3] was given based on the paid participants using headphones in sound isolated boosts, while our setup was crowd-sourced. Regarding the recommendation of about 30 listeners, [3] says (quoting) “The  above numbers  are  for  paid  participants  in  carefully controlled  conditions. In  less  controlled  scenarios,  such  as crowdsourcing ... our  advice would be to collect significantly more data and listeners”. So, there is no real contradiction. We adapted the discussion appropriately. Thank you very much for the observation!
>
> > What about using MOS for e.g. voice conversion or speech enhancement?
>
> This is a very interesting question. We added this to the discussion, specifically mentioning your examples.
>
> >  I think it would be useful to ... translate score+variance into confidence ranges
>
> This type of evaluations (confidence intervals that take into account the rater variability) are done in another our paper, “MOS vs. AB: Evaluating Text-to-Speech Systems Reliably Using Clustered Standard Errors” that has been accepted to the Interspeech conference. That paper used the intra-class correlation coefficient. Unfortunately, we could not quote it before since it also was under the blind review. We added the insights from that paper.
>
> > ... and they could give a specific recommendation (e.g. as much raters as possible? or number of ratings per rater < K (k=10?).
>
> The rater variance depend also on the voice samples themselves (see Figure 4), so the confidence intervals in the “MOS vs. AB…” paper can only be used for the analysis but not for the prediction of the number of raters. So, our recommendation is to use a limited number of tasks per rater for MOS tests. We used k=10, since we assume that the raters need some calibration. Naturally, limiting the number of ratings per rater leads to increasing the number of raters. We made it more clear in the discussion and conclusion sections.
>
> > I am not sure this is generally good recommendation (it’s for some cases, not for others: helps with fatigue, does not with respect learning the task).
>
> You’re right, this is a tradeoff issue that is hard to separate in the evaluations given the lack of the ground truth. Our goal in analyzing the second factor was to show the presence of the issues that depend on the number of tasks and not on the quality. We assume that there should be a minimal number of audio samples for the calibration, but the paper doesn’t set a goal to find this number (we’re not sure it is feasible in the current setup). Given that reducing the number of tasks per rater also requires more raters and thus reduces the impact of per-rater variability (the first factor), we do consider it beneficial. We made this part more clear in the discussion and the conclusions.
>
> >  I recommend to explain a little bit [Re clarity of the intra-class correlation]:
>
> We added the reference to the “ “MOS vs. AB: …” paper. Note also that that type of analysis helps to correctly evaluate the confidence intervals, but it cannot compensate over a small number of raters.
>
> > Why are 16+1 rater preferred to 30+1?
>
> Rephrased. The “preference” was referring to the discussion of how artificial scheduling should reflect the real world. A requirement for up to 60 samples per rater in the test of 1000 samples overall means ~17 raters in the real environment. A naive scheduling is not being able to “pack” the raters with the maximal number of samples available, so we needed to make a scheduler that makes such packing possible. Given a limited number of rater-rating pairs in the bootstrapping, we cannot reach the “perfect” packing, but we at least need to make it as packed as possible. We elaborated that in the description.
>
> > In section 3.2... please explain how samples are generated.
>
> Done
>
> > What about second test?
>
> Since this project was used solely for bootstrapping analysis, we didn’t require the raters to rate exactly N stimuli, so this information is less relevant. However, there was a typo; each sample was evaluated 10 times and not once – fixed. As for the second test, we sent the samples in the batches of 10, so having a limit of 10 samples per rater means automatically a single batch per rater. I mentioned it directly.
>
> > I don’t understand this sentence: “so, there are actually more raters participating in each simulated test than we could theoretically get in real life”
>
> This is the limitation of the bootstrapping method mentioned above. The reason is that in a simulated project, the rater pool is limited. The explanation moved to the scheduler description and removed from here in order not to confuse the reader.

---

### Decision · Program_Chairs · 2023-06-14

**Decision:**

Accept

**Comment:**

SSW2003 received 45 papers. The acceptance rate is 82%. We are pleased to inform you that your paper has been accepted by the SSW2023 Program Committee. Please read the reviews carefully and submit your camera-ready paper by June 28th. Most reviewers performed a detailed review. Please answer to their questions and consider their comments. Note that camera-ready papers are credited with one extra page to allow authors to consider reviewers’ suggestions. So max 7 pages in total including figures & refs.
The deadline for submitting the revised version (with full non-anonymized authors and refs!) is 28th June.